# Steroid Sulphatase and Its Inhibitors: Past, Present, and Future

**DOI:** 10.3390/molecules26102852

**Published:** 2021-05-11

**Authors:** Paul A. Foster

**Affiliations:** 1Institute of Metabolism and Systems Research, University of Birmingham, Birmingham B15 2TT, UK; p.a.foster@bham.ac.uk; Tel.: +44-121-414-3776; 2Centre for Endocrinology, Metabolism and Diabetes, University of Birmingham, Birmingham Health Partners, Birmingham B15 2TT, UK

**Keywords:** steroid sulphatase, cancer, sulphatase inhibitor, STX64, Irosustat, dual aromatase–sulphatase inhibitors, STX681

## Abstract

Steroid sulphatase (STS), involved in the hydrolysis of steroid sulphates, plays an important role in the formation of both active oestrogens and androgens. Since these steroids significantly impact the proliferation of both oestrogen- and androgen-dependent cancers, many research groups over the past 30 years have designed and developed STS inhibitors. One of the main contributors to this field has been Prof. Barry Potter, previously at the University of Bath and now at the University of Oxford. Upon Prof. Potter’s imminent retirement, this review takes a look back at the work on STS inhibitors and their contribution to our understanding of sulphate biology and as potential therapeutic agents in hormone-dependent disease. A number of potent STS inhibitors have now been developed, one of which, Irosustat (STX64, 667Coumate, BN83495), remains the only one to have completed phase I/II clinical trials against numerous indications (breast, prostate, endometrial). These studies have provided new insights into the origins of androgens and oestrogens in women and men. In addition to the therapeutic role of STS inhibition in breast and prostate cancer, there is now good evidence to suggest they may also provide benefits in patients with colorectal and ovarian cancer, and in treating endometriosis. To explore the potential of STS inhibitors further, a number of second- and third-generation inhibitors have been developed, together with single molecules that possess aromatase–STS inhibitory properties. The further development of potent STS inhibitors will allow their potential therapeutic value to be explored in a variety of hormone-dependent cancers and possibly other non-oncological conditions.

## 1. Introduction

The synthesis and subsequent action of oestrogens and androgens regulate many aspects of normal mammalian endocrine physiology. These sex steroids play key roles in maturation, sexual development, and reproduction. In disease, dysregulation of oestrogen or androgen synthesis are hallmarks of many pathologies, including cancers and gynaecological conditions. Indeed, sex steroids are implicated to have an impact on cardiovascular disease [1,2], cognitive health [3,4], and even colonic motility [5]. A pivotal step in the synthesis and subsequent bioavailability of active oestrogens or androgens is through the action of steroid sulphatase (STS) which hydrolyses sulphated sex steroids. Thus, over the past 30 years, the development of STS inhibitors to treat potentially many diseases has been of significant interest to many researchers. This review will explore the past, present, and future of STS inhibitor development and will in particular focus on the research by Prof. Barry Potter, who has been instrumental in the many successes in this field. Other recent reviews on this area sufficiently cover many other groups’ significant contributions to STS inhibitor chemistry and biology [6,7,8,9].

## 2. Oestrogen and Androgen Synthesis

Oestrogens and androgens are synthesised through the conversion of androstenedione (A4) and dehydroepiandrosterone (DHEA), both of which are androgen precursor steroids made in the adrenal cortex (Figure 1). To enhance solubility, the majority of DHEA is transported in the blood in its sulphated form, DHEA-sulphate (DHEA-S) which acts as a circulating reservoir for the formation of downstream active oestrogen and androgens in peripheral tissues [10]. Consequently, STS desulphation of DHEA-S to DHEA is the main route in the formation of all active oestrogens and androgens. For androgens, DHEA can be further metabolised by 3β-hydroxysteroid dehydrogenase (HSD)/isomerase to A4, which can then be converted to testosterone through 17β-HSD type-3. For oestrogens, both A4 and testosterone can be aromatised to estrone (E_1_) and oestradiol (E_2_), respectively. Further to this, relatively high circulating concentrations of estrone sulphate (E_1_S), which can be converted to E_1_ by STS, is also a route to biologically active oestrogen synthesis. Only one STS is responsible for the hydrolysis of E_1_S and DHEA-S [11]. Thus, STS activity provides key steps in regulating sex steroid bioavailability [12]. Since STS is often expressed at high levels in tissues in which hormone-dependent cancers occur, its inhibition should have potential therapeutic value.

For the last 30 years, considerable research has been carried out to design, synthesise, and test STS inhibitors. This has led to the identification of many interesting compounds which will be reviewed below. During this time, there has only been one STS inhibitor that has reached clinical trials, Irosustat (**1**, also known as 667 Coumate, BN83495, or STX64) [13,14]. This drug has been tested in phase I/II trials in various settings which are examined in this review, but mainly in patients with metastatic breast cancer. The results from these trials have been promising, but with Irosustat, still the only compound to have been clinically tested, it is now of interest to review the rationale leading to the development of this novel type of enzyme inhibitor and to assess if STS inhibitors will prove of future therapeutic value in the treatment of hormone-dependent cancers and other hormone-related conditions.

## 3. Why Develop Steroid Sulphatase Inhibitors?

Blood levels of E_1_S and DHEA-S are much higher than those of their unconjugated counterparts [12,15,16]. In the first-ever phase I trial of Irosustat, for which serum E_1_, and E_2_ and E_1_S concentrations were measured by a gas chromatographic–tandem mass spectrometry (GC–MS/MS) method, levels of E_1_S were five- to seven-fold higher than those of E_1_ but up to a 100-fold higher than the serum E_2_ concentrations [17,18]. Furthermore, due to their binding to serum albumin, steroid sulphates have a much longer half-life in blood (10–12 h), compared to unconjugated oestrogens (20–30 min) [19]. In vitro, there is strong evidence that steroid sulphates, such as E_1_S and DHEA-S, can be hydrolysed by MCF-7 breast cancer cells, LNCaP prostate cancer cells, and most other types of cancer cells [20,21]. STS activity is present at high levels in homogenates of most hormone-dependent tumours and this has led to the assumption that, by gaining access into cells, steroid sulphates will be readily hydrolysed to their unconjugated forms by STS. This enzyme is a membrane-based enzyme located on the endoplasmic reticulum [22]. Once hydrolysed, E_1_ formed from E_1_S can be reduced to E_2_ by 17β-HSD1 which is present in most hormone-dependent tumours and overexpressed in some breast [23,24,25] and endometrial cancers [26].

In addition to the formation of E_2_ from E_1_S, there has also been considerable interest, from groups developing STS inhibitors, in the role that Adiol may have in tumour development. Although an androgen, Adiol can bind to the ER, and there is good evidence that it stimulates the proliferation of hormone-dependent breast cancer cells in vitro and carcinogen-induced mammary tumours in rodents [27,28,29]. Furthermore, Adiol has recently been shown to stimulate the androgen receptor (AR) in LNCaP prostate cancer cells leading to proliferation [30]. Thus, blocking STS action may have numerous effects on the availability of various sex steroids which may prove hugely beneficial in patients with hormone-related disease.

In the context of hormone-dependent cancer, and in particular, breast cancer, targeting STS may lead to benefits for patients who have progressed or failed to respond to aromatase inhibitors. Studies have shown that once treated with aromatase inhibitors such as letrozole, breast cancer cell lines upregulate STS expression and activity [31]. Thus, it may be the case that STS expression is elevated in breast tumours from patients treated with aromatase inhibitors, although this remains to be directly tested experimentally. This suggests that upon loss of oestrogen synthesis from the androgen pathway, some hormone-dependent tumours may preferentially utilise the STS pathway to maintain E_1_, and subsequently, E_2_, generation through hydrolysis of sulphated oestrogens found in circulation.

Additionally, as regards why developing effective clinically efficacious STS inhibitors remains an important research goal, it is important to note that strategies to reduce hormone synthesis and action have been the cornerstone of many treatment regimens against endocrine disease. Thus, it is logical to assume that STS inhibition will also be an effective therapeutic option. Furthermore, many hormone-dependent cancers become resistant to current therapies. Both primary and secondary required resistance to aromatase inhibitor is relatively common and is considered a major concern for effective treatment [32]. This opens the door for complementary strategies that completely ablate hormone action and through which STS inhibition should play a clinical role. For example, combining STS inhibitors with already gold standard treatment regimens that include either oestrogen receptor α inhibitors (e.g., Tamoxifen) or aromatase inhibitors is one way that remains to be explored fully.

Once clinically available, what potential side effects would be expected in patients treated with STS inhibitors? As mentioned below (see Section 7), patient tolerance of STS inhibition has been excellent with very limited toxicologies encountered. The most common side effect seen has been dry skin [17], with some patients also experiencing nausea. The maximum tolerated dose in humans of Irosustat (and thus STS inhibition) remains to be determined, even after a dose-escalation study confirmed it was above 80 mg [33]. Long-term side effects following prolonged STS inhibition will most likely mimic what has already been observed in patients who have undergone other forms of endocrine therapy, such as ERα blockers and aromatase inhibitors. In particular, osteoporosis may occur due to the reduction in circulating oestrogens which have bone protecting effects through regulation of bone remodelling. However, these potential side effects remain to be elucidated in further clinical trials.

## 4. Steroid Sulphatase Expression and Activity

### 4.1. Breast and Gynaecological Cancers

Using either reverse transcription–polymerase chain reaction (RT–PCR), immunohistochemical (IHC), or radiolabelled assay methods several studies have examined the expression and/or activity of STS in hormone-dependent cancers. Expression of STS mRNA was found to be significantly higher in malignant breast tissue compared with normal breast tissue, consistent with the high levels of STS activity detected in cancerous breast tissues [34,35]. Furthermore, STS expression is an independent prognostic indicator in predicting relapse-free survival, with high levels of expression being associated with a poor prognosis [36,37,38,39]. However, a separate study on Norwegian patients identified elevated STS expression in breast tumours was strongly associated with a significantly lower incidence of relapse and distant metastasis leading to improved prognosis in those cohorts [40]. STS expression was also enriched in tumours overexpressing HER2, suggesting a complicated, and as yet unexplored, relationship between oestrogen availability, HER2+ cancer, and prognosis [40]. It is also of interest to note that postmenopausal breast cancer patients on neoadjuvant aromatase therapy were found to have elevated STS and 17β-HSD type-1 expression in their tumours, indicating a potential compensatory response of breast carcinoma tissue to oestrogen depletion.

High levels of STS activity have also been detected in endometrial cancer tissue with activity being 12 times higher than in the normal endometrium [41]. In an IHC study, no STS expression was detected in normal endometrium tissue although it was detected in 86% of the cases of endometrial cancer tissue [42]. However, more recent studies on patient endometrial cancer tissue have not shown changes in STS mRNA or protein expression, compared to normal endometrial controls [43,44]. For ovarian cancer, Milewich and Porter originally found that STS was present in cells derived from ovarian cancers [45]. Using IHC staining, STS was found to be present in 71% of ovarian clear cell adenomas [46]. In an examination of the ability of ovarian cancer tissues to form E_2_ from E_1_S Carlstrom et al. detected significant levels of STS activity in ovarian tumour tissues and also found a significant correlation between ovarian tumour STS activity and serum E_2_ levels [47]. STS activity was detected in 97% of ovarian cancer specimens examined [48]. Importantly, in this study, the median progression-free survival time was significantly longer in patients who had low STS activity compared with those with high activity. Indeed, patients with high-grade serous ovarian cancer expressing elevated levels of oestrogen sulphotransferase (SULT1E1), an enzyme involved with sulphating E_1_ and E_2_, thus inactivating them, had better prognostic outcomes [49]. This suggests that higher STS activity may result in poorer outcomes for these patients and implies STS inhibition may be a valid treatment option. Despite these studies, it remains unknown whether STS activity is elevated in human ovarian cancer, compared to normal ovarian tissue, with one study suggesting there is no change when examined through IHC [49].

### 4.2. Prostate Cancer

STS activity has also been detected in the prostate gland which is an important site for the peripheral formation of biologically active androgens from circulating precursors such as DHEA-S [50]. Research carried out by Labrie et al. has suggested that production of biologically active androgens, by an intracrine mechanism within the prostate, may make a similar contribution to concentrations resulting from the uptake of testosterone from the circulation [51]. IHC studies have revealed that STS is present in 85% of malignant specimens of prostate cancer tissue but absent in the non-neoplastic peripheral zones [52]. Recent evidence suggests that biologically active androgens can be formed from DHEA-S within the prostate, with STS playing the crucial initial synthesis role [53]. Indeed, there is growing compelling evidence that castration-resistant prostate cancers treated with the 17β-hydroxylase inhibitors abiraterone or enzalutamide, may utilise adrenal androgens, particularly DHEA-S, through STS action [54]. This suggests that STS inhibition may be of clinical benefit to patients on androgen deprivation therapy.

While androgens have generally been considered to be the main stimulus for the development and growth of tumours of the prostate, there is also increasing interest in the role oestrogens may have in the aetiology of this disease [55,56,57]. Indeed, elevated oestrogen receptor α (ERα) expression in prostate cancer is associated with progressive disease [58]. Uptake of E_1_S has been observed in the rat prostate, with this uptake increasing during aging suggestive of their involvement in prostate cancer development [59]. Serum E_1_S concentrations have also been found to be elevated in patients with prostate cancer, compared with those from age-matched controls, and were significantly higher in patients with a poor prognosis [60]. However, the potential role of E_1_S and subsequent desulphation by STS in prostate cancer remains speculative and will no doubt be complicated by the complex array of molecular interactions fostered by oestrogen and androgen receptors [61].

### 4.3. Colorectal Cancer

STS is also present in colon carcinomas and many colorectal cancer cell lines [62], suggesting that this cancer, which is a major cause of cancer-related deaths in both men and women, might also be a potential target for STS inhibitor therapy [63,64]. In postmenopausal women, the reduction in oestrogen levels that occurs at menopause appears to be associated with a reduction in the risk of colorectal cancer [65]. Paradoxically, epidemiological data suggest a protective effect associated with the use of hormone replacement therapy [66]. It has been suggested that these conflicting reports relate to molecular changes linked to changes in oestrogen receptor expression status in the colon during carcinogenesis [67]. Oestrogen receptor β (ERβ), the predominately pro-apoptotic oestrogen receptor, is progressively lost through colon adenoma and carcinoma development [68,69]. This may result in the dominance of ERα in utilising available E_2_ as a mitogen.

STS generates E_1_ from E_1_S and the E_1_ formed, which is only a very weak oestrogen, requires reduction to E_2_ by 17β-HSD1 to be biologically active. In the gastrointestinal tract the expression of 17β-HSD2, the enzyme that inactivates E_2_, but not 17β-HSD1, has been detected [70]. Indeed, tissue concentrations of E_1_ and E_2_ have been measured in samples of malignant and non-neoplastic colon tissues. Concentrations of these oestrogens were then related to the expression of STS and oestrogen sulfotransferase (SULT1E1), one of the sulfotransferases involved in the transformation of E_1_ to E_1_S [71]. Concentrations of E_1_, but not E_2_, were found to be higher in malignant than in non-neoplastic colon tissue. This finding is consistent with the high levels of 17β-HSD2 expression that have been detected in this tissue.

STS and SULT1E1 expression were detected in 60% and 40%, respectively, of the colon cancer samples examined, but only SULT1E1 expression was detected in non-neoplastic colonic tissues. The potential clinical significance of STS and SULT1E1 in colon cancer tissue was also examined by Sato et al. in an IHC study of 328 patients with this cancer. Patients whose tumours did not express STS but did express SULT1E1 had a better clinical outcome than those whose carcinomas expressed STS. Whilst STS expression itself was not prognostic, the ratio of STS: EST expression was an independent prognostic factor for overall survival. Further work in this area has shown STS activity is elevated in human colorectal cancer tumours, compared to match normal tissue, and that increased STS activity is associated with greater colorectal cancer cell growth [72]. Importantly, inhibition of STS with STX64 blocked colorectal cancer proliferation in mice. It would therefore appear that the potential role of STS inhibitor therapy for patients whose colon carcinomas expressing STS is worthy of further investigation.

## 5. The Development of STS Inhibitors

A variety of approaches have been used to design STS inhibitors with resultant compounds generally falling into three categories: alternative substrates (including competitive reversible inhibitors), reversible inhibitors, and irreversible inhibitors. The very first class of STS inhibitor, an alternative substrate, was a series of 2-(hydroxyphenyl) indole sulphates, one of which had an IC_50_ of 80 μM [73]. In subsequent years, many STS inhibitors from synthetic and naturally occurring steroid-related compounds have been generated, the most potent, at 2 μM, being 5-androstene-3β, 17β-diol-3-sulphate (Figure 2, compound **1**), [74]. However, a significant limitation for these alternative substrates is they could potentially be metabolised in vivo and form oestrogenic molecules with ER binding affinity. This would likely make them mitogenic factors and of limited clinical benefit for the treatment of hormone-dependent disease.

Thus, during the 1990s, efforts by various research groups aimed to design and develop potent, reversible STS inhibitors. Leading many of these novel developments in STS inhibitor chemistry was Prof. Barry Potter, then at the University of Bath, UK. Studies initially focused on replacing the sulphate group (OSO3^-^) of E_1_S with surrogates or mimics such as phosphonates [75,76,77], sulfonates [78], sulfonyl halides [79], sulphonamide [80], and the methylenesulfonyl group [81,82]. Made to compete against E_1_S for the STS enzyme active site, these compounds were designed to remain metabolically stable and thus not to act as substrates.

The first compound to be specifically synthesised was E_1_-MTP (Figure 2, Compound **2**) and from this, a series of related STS surrogates was designed which included estrone-3-O-sulfamate (Figure 2, Compound **3**, EMATE). This compound possesses the active pharmacophore, i.e., an aryl sulfamate ester, required for potent STS inhibition [82,83]. Unfortunately, EMATE did not progress into clinical trials for breast cancer since it was discovered, unexpectedly, that it was a ‘super oestrogen’ being five times more active than ethinyloestradiol when administered orally to rodents [84]. It is now known that compounds with an aryl sulfamate ring, such as EMATE and STX64, are sequestered into erythrocytes where they bind reversibly to carbonic anhydrase II [85,86]. Therefore, these drugs are able to avoid the first-pass inactivation in the liver. The oestradiol analogue of EMATE, estradiol-3-O-sulfamate did undergo phase I/II trials for hormone replacement therapy but proved to be devoid of oestrogenicity in postmenopausal women.

Despite EMATE’s failure to reach the clinic, its discovery stimulated the design of a range of non-oestrogenic STS inhibitors with equipotency. Several groups examined a number of 1–3 ringed non-steroid-based sulfamates [87,88]. There have also been some modifications made to the A and D rings of the steroid oestrane nucleus to limit the oestrogenicity of these inhibitors [89,90]. Importantly, compounds shown to have similar potency to EMATE incorporate the active pharmacophore for STS inhibition, i.e., an aryl sulfamate ester [91].

Further attempts to create compounds devoid of oestrogen action resulted in the discovery of non-steroidal sulfamates as STS inhibitors. A coumarin sulfamates series was made as part of this development programme, one of which, 4-methylcoumarin-7-O-sulfamate (Figure 2, Compound **4**, COUMATE), inhibited STS activity in MCF-7 breast cancer cells by >90% at 10 μM [89]. Continued research on a tricyclic coumarin sulfamates series identified 667COUMATE (Figure 2, compound **5**, also known as BN83495, STX64, or Irosustat) as more potent than EMATE, possessing an IC_50_ of 8 nM in placental microsomes [92]. Importantly, 667COUMATE did not induce oestrogenic responses when examined in vitro and in vivo [93,94].

## 6. Next Generation Steroid Sulphatase Inhibitors

STX213 (Figure 2, Compound **6**), a steroid-based N-propyl piperidinedione derivative of EMATE, is a second-generation STS inhibitor developed in the early 2000s. In placental microsomes, STX213 had an IC_50_ value of 1 nM, 18-fold more potent than EMATE [95]. The rat uterotrophic assay confirmed that in vivo STX213 was devoid of oestrogenicity [96]. Although results from in vitro studies suggested STX213 was approximately threefold more potent than Irosustat (STX64), it proved to have a considerably longer duration of STS inhibition in rodents [97]. After administration of a single oral dose of Irosustat (10 mg/kg) rat liver STS recovered by 50% by day 4. In contrast, a single oral dose of STX213 (10 mg/kg) 50% restoration of liver STS activity was not achieved until after day 12. It is believed the turnover time for STS is approximately 3–5 days; thus, the prolonged activity of STX213 suggests it is sequestered into, and subsequently released from, peripheral tissues.

An ovariectomised nude mouse model was developed to compare the in vivo efficacies of Irosustat and STX213 [97]. For this, mice were inoculated with MCF-7 cells (MCF-7_WT_) or MCF-7 cells overexpressing STS (MCF-7_STS_). Overexpression of STS is a better representation of what is seen clinically in breast tumours. Growth of MCF-7_WT_ and MCF-7_STS_ xenograft tumours was reduced by both Irosustat and STX213, although dosing with STX213 resulted in a much greater degree of tumour growth inhibition than that with Irosustat. Further studies on a third-generation STS inhibitor, STX1938 (Figure 2, Compound **7**), showed it also significantly reduced MFC-7_WT_ and MCF-7_STS_ xenograft growth in vivo [98] and provides further proof of concept for STS inhibition in treating hormone-dependent breast cancer.

In light of these encouraging results obtained with STS inhibition blocking xenograft tumour growth derived from breast cancer cells, further studies examined their efficacy in a nude mouse model of hormone-dependent endometrial cancer [99]. At 1 mg/kg per day, while Irosustat inhibited xenograft growth by 50%, STX213 reduced tumour growth to a much greater extent (67%). Weekly dosing scheduled at 1 mg/kg per day was also tested, but only STX213 was effective. When given weekly, Irosustat did not completely block liver STS activity and failed to reduce plasma E_2_ concentrations. These results indicate that for maximal efficacy, when used for the treatment of hormone-dependent cancers, it will be essential to ensure that total inhibition of STS activity is achieved.

## 7. STS Inhibitors in Clinical Trials against Hormone-Dependent Breast Cancer

Despite animal studies on next-generation STS inhibitors demonstrating their superior inhibitory attributes, all subsequent clinical trials of STS inhibition have been performed using Irosustat (STX64).

The first phase I clinical trial of an STS inhibitor took place during the mid-2000s in postmenopausal patients with advanced breast cancer. Irosustat was given orally to 14 women (nine patients at 5 mg and five at 20 mg dose) as an initial dose followed a week later by three biweekly cycles of 5-day dosing and 9 days off treatment. Results were very encouraging, with Irosustat well tolerated and four patients, all of whom had progressed on aromatase inhibitors, exhibiting stable disease for 2.75 to 7 months. Patient STS activity, as measured in peripheral blood lymphocytes (PBLs), was suppressed by >95% at the 5 mg/day and 20 mg/day doses tested [17]. This degree of STS inhibition is associated with moderate but significant reductions in the median concentrations of E_1_ (57–76%), E_2_ (38–39%), and testosterone (27–30%). Furthermore, the median concentration of Adiol, a steroid that possesses some oestrogenic activity, decreased by 70–74%. Unexpectedly, serum Adione median concentrations also fell by 62–72%, suggesting that, at least in postmenopausal women, a significant proportion of this steroid is derived from the peripheral conversion of DHEA-S. These results show that while median serum concentrations of Adiol, Adione, and E_1_ all decreased by approximately 70%, the reductions for testosterone and E_2_ were less, at about 30%.

Another phase I clinical trial with Irosustat aimed to identify the optimal dose for STS inhibition in ER+ breast cancer patients [33]. A single oral dose was given, followed by a 7-day observation period. After this, a daily dose of the drug was given for 28 days with an extension phase, in which dosing was only continued if a patient benefit was observed. Five different dose concentrations of Irosustat were examined, with the highest being at 80 mg. Results were highly encouraging, with all patients in the 5, 20, 40, and 80 mg cohorts achieving ≥95% STS inhibition in peripheral blood mononuclear cells after 28 days of treatment. The maximum tolerated dose was not reached, and the 40 mg dose was calculated to be optimal. Aside from determining this optimal dose of Irosustat, this clinical trial also reported that five patients (out of 50) were disease progression-free for at least 24 weeks. Although the study did not test drug efficacy, these results are particularly impressive considering these patients had been heavily pre-treated with standard therapeutic regimes prior to joining the trial.

Further clinical trials of Irosustat have more recently been conducted, examining STS inhibition in treatment naïve postmenopausal patients, as opposed to heavily pre-treated patients with advanced disease. The ‘IPET’ clinical trial took a ‘window of opportunity’ to treat hormone-dependent early breast cancer patients with Irosustat prior to surgery [100]. Thus, this trial examined Irosustat’s effects as a single agent in a patient with breast cancer. This study was designed to examine how STS inhibition impacted tumour cell proliferation, as determined by PET scanning measure of cellular uptake of 3′-deoxy-3′-[18F] fluorothymidine (FLT) and Ki67 staining. A relatively modest cohort of 13 women was recruited to the study and given a dose of 40 mg daily for up to 2 weeks. For Ki67 staining, six out of seven patients exhibited a reduction in immunoreactivity, suggesting a reduced proliferative rate. FLT uptake was also significantly reduced (a < 20% reduction) in these patients. In general, Irosustat was well tolerated. Intriguingly, STS inhibition also lowered aromatase and 17β-HSD type-1 expression in the tumours, the mechanism of which remains unknown but may suggest hormonal regulation of these enzymes in cancer. Importantly, these data are the first to show STS inhibition may have clinical benefit in patients with early hormone-dependent breast cancer, albeit in a small cohort of patients. In order to boost recruitment, a multicentre trial is now required to finally determine how beneficial Irosustat could be in patients with early ER+ breast cancer.

Another suggested strategy for using STS inhibitors clinically has been to combine them with aromatase inhibitors, which have already shown considerable success in treating patients with ER+ breast cancer. The idea of developing compounds with dual aromatase–sulphatase inhibition has been around for over 15 years and has been explored in detail (see below). However, with regard to the clinical application of this theory, another recent phase II trial, the ‘IRIS’ study, examined this premise directly [101]. This multicentre, open-label trial treated advanced breast cancer patients with Irosustat (oral, 40 mg/day) combined with first-line aromatase inhibitors to assess evident clinical benefit in attempts to further ablate circulating E_2_ concentrations. Furthermore, the safety profile of combining an STS inhibitor and aromatase inhibitor was also determined. In addition, it was believed that combining STS and aromatase inhibition would further ablate available circulating hormones and provide the greatest benefit to patients. In total, 27 women were recruited for the study, although 4 discontinued. The study reports a clinical benefit rate of 18.5%, and of the five patients that showed a response, the median duration was for 9.4 months and the median progression-free survival was 2.7 months. The treatments were well tolerated. Overall, the study confirms the scientific merit of administering both STS and aromatase inhibitors for ER+ breast cancer patients and future clinical trials should be performed to further evaluate this strategy moving forward.

## 8. Other STS Inhibitor Clinical Trials

The most advanced STS inhibitor clinical trials have involved patients with breast cancer (see above). Since oestrogen ablation through STS inhibition may have benefits against other malignancies, other trials have occurred with some success. A review of these trials in other cancers and endocrine conditions has been recently comprehensively presented [8].

Briefly, early pre-clinical research identified the potential for STS inhibitors to be a successful strategy against oestrogen responsive endometrial cancer [99]. These data supported the development of a clinical trial to examine STS inhibition against standard care in patients with endometrial cancer. Patients with advanced/metastatic or recurrent endometrial cancer were recruited across 11 European countries in a randomised, two-arm study [102]. Irosustat was administered orally at the optimal dose of 40 mg/day and was compared against the current standard of care, megestrol acetate (MA; 160 mg/day orally). A total of 71 postmenopausal patients were recruited onto the study with confirmed oestrogen or progesterone positive endometrial tumours. Irosustat again demonstrated good tolerance in patients. Of those treated with Irosustat, 36% were alive after 6 months, with 47% of these showing stable disease, compared to 32% in the MA treated cohort. Unfortunately, and after futility analysis of the two cohorts, the trial was terminated early as there was no significant difference in response rates and survival, with median progression-free survivals at 16 weeks for Irosustat patients, compared to 40 weeks for MA treated patients. Interestingly, since this trial was performed, evidence suggests that STS expression in endometrial cancer is not a prognostic factor for outcomes [103]. However, only 27% of patients were STS positive, implying that stratification of patient population may be required to show STS effect in endometrial cancer. Overall, and despite these potentially unfavourable data for STS inhibition in endometrial cancer, this trial did demonstrate STS inhibition as effective and well tolerated and thus provides further support for the rationale behind the development of these compounds.

Another trial of Irosustat determined its potential effect in treating patients with hormone-dependent prostate cancer. It has been known for some time that STS activity is present in human prostate cell lines [104] and prostate cancer tissue [52]. Indeed, over-expression of STS in prostate cancer may be associated with inducing Wnt/beta-catenin signalling and the upregulation of Twist1 and Hif-α [105], both pathways involved in cancer migration. 

In prostate tissue and cancer, it is hypothesised that the STS pathway can generate the precursor androgens (DHEA from DHEA-S) necessary to facilitate the intracrine synthesis of active androgens [106,107,108]. Recently, there has been direct evidence for this pathway being active in the human prostate, with STS activity elevated in castration-resistant prostate cancer [54]. This study demonstrated that intracrine androgen synthesis was regulated through STS activity, and that suppression of STS could be a novel additional treatment for prostate cancer. Since DHEA-S concentrations are present in high levels in most men, this pathway may remain an important mitogenic route for hormone-dependent prostate cancer and castration-resistant malignancy. A clinical trial, which remains to be fully reported, has examined how Irosustat impacted hormone concentrations in patients suffering from castration-resistant prostate cancer who were being treated with anti-androgen therapy through a US phase I dose-escalation study. The aims were to assess safety, tolerability, pharmacokinetics, and a selection of circulating hormone concentrations of STS inhibition in men, and this was the first time such a therapy had been used in males. Results indicated Irosustat had a good safety profile with expected pharmacokinetic readouts. Unsurprisingly, plasma concentrations on non-sulphated androgens (androstenediol, testosterone, and DHEA) were all suppressed in all doses tested (20, 40, and 60 mg). The DHEA:DHEA-S ratio was significantly decreased, indicating the importance of STS in biological availability of androgen precursors. Further data on patient outcomes from this clinical trial remain to be reported. However, these initial results are encouraging in supporting a potential role for STS in castration-resistant prostate cancer through ablation of available androgen precursors for subsequent downstream intracrine androgen synthesis.

## 9. The Further Development of STS Inhibitors

The next stages of STS inhibitor development aimed to improve understanding of the structure–activity relationship (SAR) of these compounds. Initial research focused on derivatives of oestrone sulfamate (EMATE), a potent irreversible STS inhibitor with this compound substituted at the 2- and/or 4-positions of the A-ring with a nitro group, halogens, alkyl groups, and a cyano group. The D-ring was modified by the removal of the C17 carbonyl group [109]. Compounds were tested against placental microsomes and MCF-7 breast cancer STS activities. Results indicated that these EMATE derivatives with A-ring electron-withdrawing substituents (4-nitro, 2-halogens, and 2-cyano) exhibit a greater STS inhibitory activity. However, compounds with a nitro group at the 2-position lend themselves to a significantly lower STS potency.

More extensive SAR work, now looking at Irosustat chemistry, built on early basic studies on tricyclic coumarin-based sulfamates [13] which centred around contraction and expansion of the aliphatic ring. Subsequent studies designed modifications that further expanded the aliphatic ring leading to significant increases in STS inhibitory potency from 7 to 11 members, with this effect reduced on greater expansions [110]. The most potent STS inhibitors possessed IC_50_s between 0.015 and 0.025 nM. Other modifications made to Irosustat, such as *N*,*N*-dimethylation of the sulfamate group or relocation of the sulfamate group to another position, either significantly reduced or abolished STS inhibitory effects.

More recently, arylamide derivatives with terminal sulfonate or sulfamate moieties have been designed and tested against STS activity. The structures of these novel compounds possessed different pharmacophore regions to include sulfonate, sulfamate, and N-substituted sulfamate groups. Furthermore, the size of the aliphatic ring was also altered to contain five or six-membered rings [111]. Unsurprisingly, the most potent derivative contained the sulfamate group showing an IC_50_ concentration of 0.42 μM against STS activity in JEG-3 cells (Figure 3, Compound **1**). Additionally, the cyclohexyl motif was shown to have a more favourable STS inhibitory activity, compared to the cyclopentyl. Further work on this aryl sulfamate moiety has recently involved replacing the cycloheptxyl ring with smaller (cyclopentyl) or larger (cycloheptyl or adamantly) rings, or by replacing it with aryl rings [112]. From these series, the most active compounds were adamantly derivatives as they are thought to mimic the five-membered D-ring of estrone sulfate and the seven-membered ring of Irosustat. The most potent adamantly derivative possessed an IC_50_ of 14 nM when tested in STS activity in JEG-3 cells (Figure 3, Compound **2**). This compound also showed strong anti-proliferative properties when tested against hormone-dependent breast cancer cell line T47D, with an IC_50_ of around 1–6 μM. These compounds, therefore, show some promise as new routes to synthesis more potent STS inhibitors and will require future lead optimisation prior to in vivo testing.

Another group has recently identified a series of 4-(piperazinocarbonyl)-aminosulfamates showing potential STS inhibitory activity [113]. Their strategy focused on synthesising derivatives containing fluorine or chlorine within the aryl-sulfamate pharmacophore. The resultant compounds had high STS inhibitory potency of IC_50_ at 5.1 (Figure 3, Compound **3**) and 8.8 nM (Figure 3, Compound **4**) in JEG-3 cells. Other attempts to optimise these compounds further did not achieve an increase in potency.

The above look at the development of STS inhibitors has primarily focused on work emanating from Prof. Barry Potter’s laboratory. However, this is by no means the complete picture. Many other groups have designed and tested STS inhibitors [6,114,115,116,117] and will not be reviewed here.

## 10. Dual Aromatase–Sulphatase Inhibitors (DASI)

Within this review, it has been shown that both the aromatase and STS routes synthesise oestrogens (see Figure 1). Clinical trials are now demonstrating STS inhibition as clinically effective in patients with hormone-dependent breast cancer, and thus, it would be significant to combine STS inhibition with aromatase inhibitors. This strategy should, in theory, limit estrone synthesis and steroids such as Adiol, and enhance response rates to endocrine inhibitor therapy. Indeed, this strategy has been tested clinically, as mentioned in the above clinical trial section through the ‘IRIS’ trial [101], and has shown some success. However, in that trial, Irosustat was administered with an aromatase inhibitor. It would be advantageous to develop a single molecule with dual aromatase–sulphatase inhibitor (DASI) properties. Evidence suggests there are clinical benefits to using a single drug, compared to multiple drugs [118], particularly when considering toxicity. A single drug approach would avoid drug–drug interaction and lead to more straightforward pre-clinical efficacious dose testing. Furthermore, since tumours may develop resistance to single-targeted drugs, hitting a second target may overcome or circumnavigate that resistance.

The initial design of an active DASI compound took advantage of the fact that certain flavonoids have aromatase inhibitory activity [119,120]. Thus, sulphamoylation of these compounds should result in molecules with DASI activity. Sulphamoylation of 4′-hydroxy and 4′,7-dihydroxyisoflavone to give 4′-mono- and 4′,7-bis-sulfamates demonstrated these compounds had STS inhibitory activity in vitro and in vivo. Unfortunately, both molecules lacked potency when compared to EMATE [121]. However, these results did confirm that DASI development was possible through altering known aromatase inhibitors.

Subsequent studies have sulphamoylated third-generation, nonsteroidal, aromatase inhibitors such as letrozole and anastrozole [87,122]. These molecules are designed with a triazole ring that coordinates reversibly to the heme iron of aromatase. Consequently, these aromatase inhibitors are reversible, which is in contrast to the irreversible steroid-based inhibitors, such as exemestane. This means that by incorporating into such molecules a phenol sulfamate ester, the STS inhibition pharmacophore, these compounds would be irreversible STS inhibitors but with reversible aromatase properties.

The first DASI synthesised came from altering YM 511 (Figure 4, Compound **1**), a very potent and selective aromatase inhibitor [123,124]. YM 511 was shown to have an IC_50_ of 0.5 nM against aromatase activity but was inactive against STS [122]. Following sulphamoylation to generate a *p*-sulfamoyloxybenzyl derivative of YM 511, impressively increased STS inhibitory properties (IC_50_ = 227 nM) but reduced aromatase inhibition (IC_50_ = 100 nM). An *m*-bromo derivative significantly elevated both aromatase and STS inhibitory activity (IC_50_ values: 0.82 nM and 39 nM, respectively). The use of the pregnant mares serum gonadotrophin (PMSG)-stimulated ovarian aromatase rat model demonstrated this molecule blocked 85% aromatase activity and 72% STS liver activity after 24 h [125]. These initial studies show the DASI concept could result in considerable therapeutic improvements for the treatment of hormone-dependent cancers and other hormone-dependent conditions.

Subsequent development of DASI compounds has identified a *p*-sulphamoylated YM 511 series through introducing substituents considered to be electron donating and/or electron withdrawing at positions ortho to the sulfamate group [126]. The *m*-sulphamoylated series of compounds has yielded the most impressive derivatives containing a substituent at the para position of the phenyl ring. One compound, STX681, an *m*-sulfamate derivative (Figure 4, compound **2**) inhibited aromatase and sulphatase activity by 82% and 98%, respectively, when given orally at 10 mg/kg to female rats [127]. Since STX681 showed promising efficacy against STS and aromatase, it was moved forward into pre-clinical in vivo models on hormone-dependent breast cancer. These models consisted of using MCF-7 cells stably overexpressing either STS (MCF-7_STS_) or aromatase (MCF-7_AROM_), both of which had been used in previous in vivo xenograft studies to investigate the efficacy of STS inhibitors [94] and aromatase inhibitors [128], respectively.

In immunocompromised ovariectomised mice, xenografts of either MCF-7_STS_ or MCF-7_AROM_ were allowed to develop. Animals with MCF-7_STS_ xenografts were given daily supplements of oestradiol sulphate (E_2_S) as a substrate for STS to generate biologically active oestradiol (E_2_) in the tumour [127]. Oral administration of either Irosustat (STX64) or STX681 completely blocked E_2_S-stimulated proliferation of tumours. Letrozole, an aromatase inhibitor, failed to stop this growth, demonstrating the importance of the STS pathway in this model and the efficacy of STX681 against the STS enzyme. In contrast, animals with MCF-7_AROM_ xenografts were given daily supplements of androstenedione (A4) as a substrate for aromatase to generate E_2_ in the tumour. In this model, oral administration of letrozole and STX681 both inhibited xenograft growth. However, Irosustat failed to block tumour proliferation in this model demonstrating the importance of aromatase in A4 to E_2_ synthesis in MCF-7 proliferation. Ultimately, these studies clearly showed STX681 was effective at inhibiting both STS and aromatase in vivo.

Since this first pre-clinical study on STX681, there has remained an interest in further developing compounds with more potent DASI properties. A group at Tohoku Pharmaceutical University in Japan synthesised and tested 4-(*p*-sulphamoylphenyl) androstenedione and 6α-*p*-sulphamoylphenyl analogues as STS and aromatase inhibitors [129]. The *p*-sulphamoylphenyl compounds were impressive aromatase inhibitors with K(i) values between 30 (Figure 4, Compound **3**) and 97 nM. They also discovered that 6α-*p*-hydroxyphenyl compounds generated from their respective sulphamoylphenyl compounds by the action of STS also possessed high K(i) concentrations (from 23 (Figure 4, Compound **4**) to 75 nM). However, these compounds did not perform well when tested against STS activity, with IC_25_s greater than 200 μM.

The majority of potent DASI compounds continue to be synthesised by the group of Prof. Barry Potter at the University of Oxford. Over the past decade, they have generated many interesting molecules with DASI properties. Back in 2010, the first study was published on impressively potent DASIs using a biphenyl core [130]. STX1983, the most active derivative (Figure 4, Compound **5**), demonstrated IC_50_ of 5.5 nM and 0.5 nM against STS and aromatase activity, respectively. Further research using a ‘merged pharmacophore’ strategy (i.e., combining the heterocyclic CYP19 binding motif of aromatase inhibitors with the aryl sulfamate pharmacophore of STS inhibitors) has produced highly potent compounds with even picomolar activity. One of the best compounds exhibits an IC_50_ of 830 pM against STS activity and 15 pM against aromatase activity using a JEG-3 assay [110]. This strategic concept of combined pharmacophores and the potential wider implications are discussed in detail elsewhere [131].

The same group went on to investigate further novel DASI compounds based on the STX681 structure. They examined a large range of modifications including relocation and replacement of the halogen atom, replacement of the methylene linker with a difluoromethylene motif, replacement of a *p*-cyano-phenyl ring with other ring structures, replacement of the triazolyl group with an imidazolyl group, and the introduction of more halogens [131]. The best molecule synthesised was a fluorinated compound composing of an imidazole ring with IC_50_ of 2.5 nM and 0.2 nM against STS and aromatase, respectively. Other novel DASI compounds included sulphamoylated letrozole [132,133,134] and anastrozole [126].

## 11. Future Outlook for STS Inhibitors

There remains much to be performed within this field, most notably pre-clinical testing of recently identified STS inhibitors and DASIs and further clinical testing of more established candidates. Most clinical trials have employed the use of Irosustat [135] and only a handful of other STS inhibitors have reached pre-clinical evaluation for the treatment of cancer [97,98,99,108]. The concept of STS inhibition in treating hormone-dependent conditions remains to be fully tested. Breast cancer trials have shown much promise, and recent evidence suggests re-evaluating STS inhibitors for the treatment of abiraterone-resistant prostate cancer [54]. No clinical trials have been conducted to examine STS inhibition in colorectal cancer despite promising pre-clinical studies in this area [62,72]

Other conditions have also not been fully explored with regard to STS inhibition. Endometriosis, where hormone-responsive endometrial tissue proliferates outside the uterus, is potentially responsive to STS inhibition [136,137]. There is a strong expression of STS in human endometrial tissue [138] and in ectopic tissue from patients with endometriosis [139]. Indeed, a higher STS activity in human ectopic endometrial tissue correlates with worse severity of disease [140]. Furthermore, inhibition of STS has been shown to decrease the size of human endometrial explants tissue that has been implanted in mice as a model of endometriosis [141]. Consequently, it is disappointing that STS inhibition has not been clinically trialled as a treatment for endometriosis, a condition that affects nearly 1 in every 10 women.

Another recent and exciting development of potential future avenues for STS inhibition lies within treating Alzheimer’s disease. There has been much speculation on the potential role of the neuroactive steroids DHEA and DHEA-S in the formation of amyloid β plaques in the brain. In the brains of patients with Alzheimer’s, DHEA concentrations are known to be increased, and DHEA-S concentrations decreased [142]. One study has tried to restore the balance of DHEA to DHEA-S by using DU-14, an STS inhibitor [143]. Among other effects, blocking STS action in rats inhibited amyloid β-induced cognitive defects in the memory of rats. Results indicated that STS inhibition may be neuroprotective against neurotoxic amyloid β accumulations in the brain and that this is possibly due to increased DHEA-S availability. Recently, these findings have been supported by a study using *Caenorhabditis elegans* who have had their STS activity (sul-2) deleted [144]. Loss of steroid sulfation led to a rise in sulphated steroids, increased life span, and inhibited protein aggregation diseases. STS inhibition with Irosustat in wild-type *C. elegans* mimicked the sul-2 deletion effects. Remarkably, Irosustat was also able to reduce Alzheimer’s disease outcomes in a rodent model of the condition. These data, therefore, suggest that STS inhibition may be beneficial in diseases associated with protein aggregation, although this remains to be clinically tested.

There are also opportunities to develop dual inhibitor strategies further which incorporate STS inhibition combined with hitting other targets. Various groups have developed molecules that possess both STS inhibition and oestrogen receptor modulating properties [145,146,147], although none of these compounds have been tested clinically. Furthermore, other groups have identified compounds targeting both STS and 17β-HSD-1 with the aim of developing these for many oestrogen-driven conditions [148]. Again, this therapeutic strategy remains to be tested in patients.

## 12. Final Remarks

Over the past 30 years, there has been a growing recognition of the importance of STS action in many hormone-dependent conditions. Thus, and considering the role STS has on the synthesis of both androgens and oestrogens, the search for potent STS inhibitors has been and remains a fruitful and interesting endeavour for many research groups. It is evident from the results of the now many clinical trials of Irosustat that this is an effective, and relatively safe, STS inhibitor in humans. From the first trial of Irosustat in postmenopausal women, it was effective both at blocking STS action in circulating peripheral blood lymphocytes and in tumour biopsies. STS inhibition also lowered plasma E_1_, E_2_, and Adiol concentrations, suggesting the importance of this enzyme in regulating blood steroid levels. However, the reductions in these steroids were modest, and it remains to be examined whether more potent STS inhibitors may further reduce the circulating availability of these steroids.

There remains significant potential for the development of DASIs, and some of these compounds are at an advanced stage of development. In particular, STX681 has proven to be an effective inhibitor of both STS and aromatase action in vivo and to have shown anti-proliferative effects against hormone-dependent breast cancer in mice. Since there remains considerable expense to develop DASI compounds further, it is encouraging that the recent IRIS clinical trial demonstrating that using both aromatase inhibitors and Irosustat showed some clinical benefit to patients with hormone-responsive breast cancer. This suggests that there are advantages to developing dual-acting inhibitor compounds for cancer therapy.

## Figures and Tables

**Figure 1 molecules-26-02852-f001:**
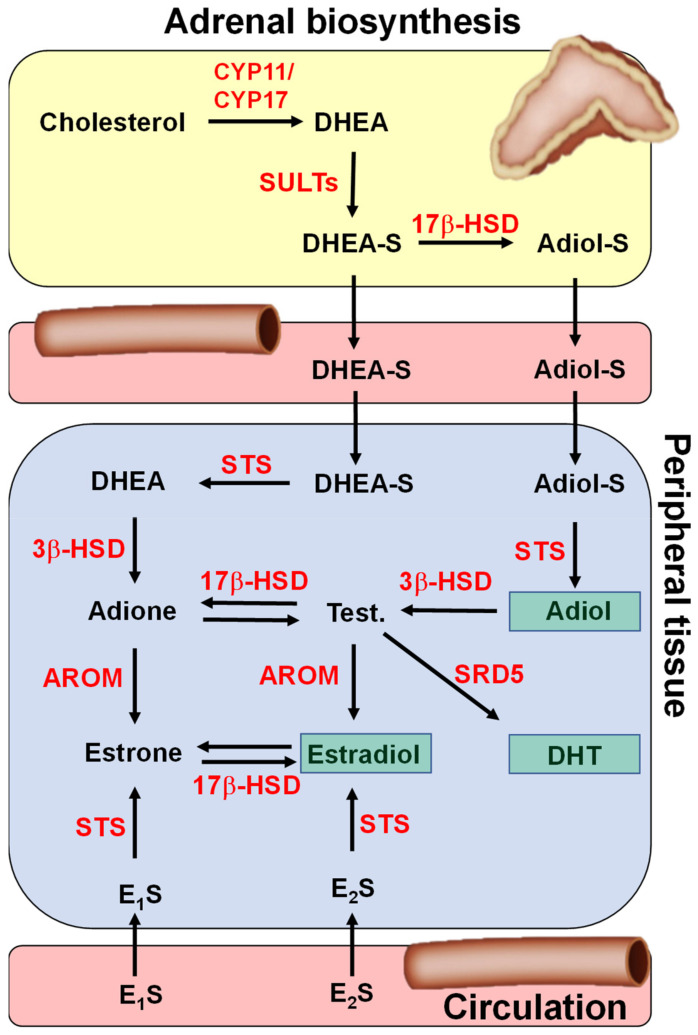
Synthesis pathways for steroids with oestrogenic and androgenic properties. Sex steroid pre-cursors, such as dehydroepiandrosterone sulphate (DHEA-S) and androstenediol sulphate (Adiol-S), are synthesised in the adrenal gland from cholesterol. The sulphate group increases water solubility, allowing them to be more readily transported in the blood. Once at target peripheral tissue, sulphated sex steroids are taken up via organic anion transporting polypeptides (not shown). Steroid sulphatase hydrolysis these sulphated steroids, allowing them to act either at oestrogen or androgen receptors or to be metabolised further to more active steroids. Additionally, in circulation and synthesised in other tissues (e.g., ovaries, adipose), E_1_S and E_2_S are available for peripheral tissue desulphation. In the diagram, the most potent steroids are shown in green boxes, although other steroids also have some oestrogenic (oestrone) and androgenic (testosterone) action. Test. = testosterone; 17β-HSD = 17β-hydroxysteroid dehydrogenase; 3β-HSD = 3β-hydroxysteroid dehydrogenases; AROM = aromatase; SRD5 = 5α-reductase type-1; E_1_S = oestrone sulphate; E_2_S = oestradiol sulphate.

**Figure 2 molecules-26-02852-f002:**
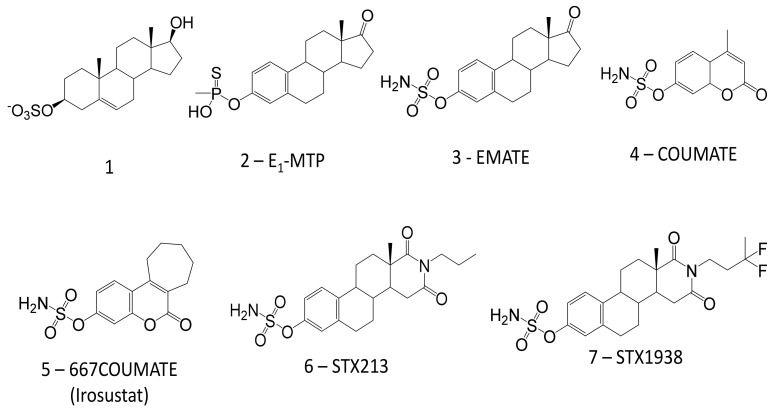
The structure of selected STS inhibitors.

**Figure 3 molecules-26-02852-f003:**
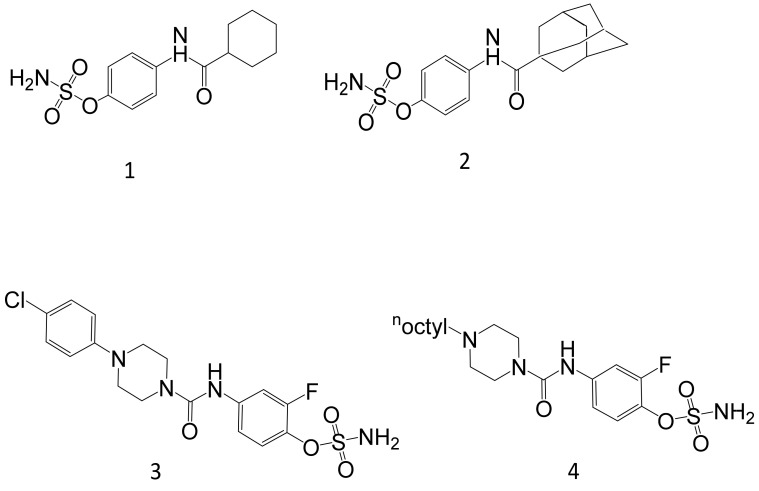
The structure of more recently identified STS inhibitors.

**Figure 4 molecules-26-02852-f004:**
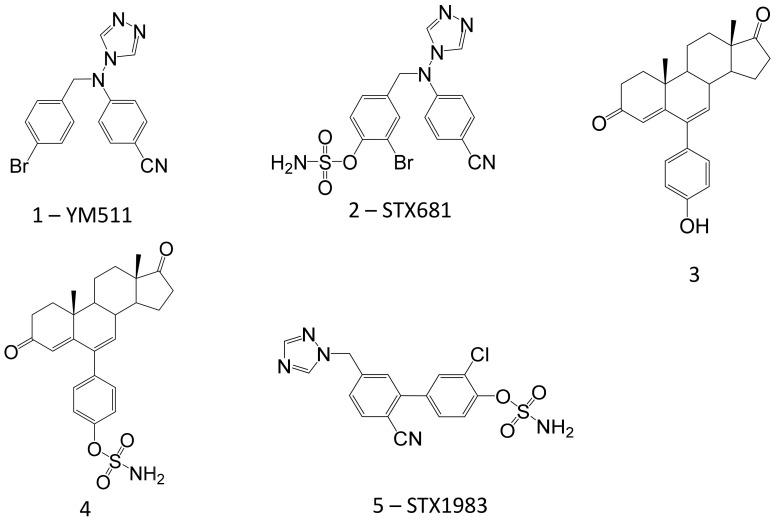
The structure of dual aromatase–sulphatase inhibitors (DASI).

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
