# Peer review of "Steroid Sulphatase and Its Inhibitors: Past, Present, and Future"

_molecules, 2021, doi:10.3390/molecules26102852_

Round 1

Reviewer 1 Report

The review paper "Steroid Sulfatase and its Inhibitors: Past, Present, and Future" is dedicated to overview of the history and current status of STS inhibitors. The paper follows particularly the work of Prof. Barry Potter, a more significant contributor in the field. In principle, the manuscript is overall well-written and organised and, especially, through putting everything in historical perspective, has an educational value. Certain issues, however, must be addressed before this manuscript can be published.

Major concerns:
1) Steroid synthesis pathway on Figure 1 must be improved, especially given that the paper is likely to be read by people with lesser or no expertise in the field. 
    - author speaks a lot of E1 and E1S throughout the text, but E1S to E1 transition is not shown and estrone abbreviation is omitted
    - the scheme should show Adiol to Testosterone conversion by 3beta-HSD, otherwise it might be misleading 
2) The manuscript section "Why develop STS inhibitors" is very important, but does not answer the question to expected depth. It should surely discuss not only purely mechanistic grounds for STS efficacy, but also put it in context of existing treatments for hormone-sensitive tumors. For instance, some discussion on their compatibility with aromatase inhibitors (lines 460-463) should be moved here and expanded. Also, it is important to compare expected adverse effects of STS inhibition with others.
3) in the section 7, it would be important to elaborate why STS did not progress as a single agent and went in the Phase II in combination only. 

Minor concerns:
- compound structures are drawn or copied carelessly, with different and frequently unreadable font size, different line thickness and structure size. Should be redrawn and unified. 
- lines 561-565: it is unclear what is meant by "all but Irosustat", probably author meant inverse, but it would be also wrong since there were some other STS inhibitors in Phase I (the cited paper is very old).
- throughout the text all the greek symbols seen as some strange symbol, alpha sometimes appears as just "a", e.g. in 5a-reductase type-1
- Figure 1 legend: organic anion polypeptides should be  "organic anion transporting polypeptides"

Some noticed typos:

145 "examine" - examined

265 "after" is missing

323 "of" is missing

385 "beta catenin" - beta-catenin

590: "wildtype c. elegans" should be wild type C. elegans (in italics)

589: "life-span" - life span

Author Response

I would like to thank the reviewer for their useful comments and their suggestions on improving this manuscript. Hopefully I have now addressed these concerns which I have outlined below.

Major concerns:
1) Steroid synthesis pathway on Figure 1 must be improved, especially given that the paper is likely to be read by people with lesser or no expertise in the field. 
    - author speaks a lot of E1 and E1S throughout the text, but E1S to E1 transition is not shown and estrone abbreviation is omitted
    - the scheme should show Adiol to Testosterone conversion by 3beta-HSD, otherwise it might be misleading

I have redrawn figure 1 to include these suggestions. E1 and E1S are now in the diagram, as is Adiol to T conversion by 3beta-HSD.

2) The manuscript section "Why develop STS inhibitors" is very important, but does not answer the question to expected depth. It should surely discuss not only purely mechanistic grounds for STS efficacy, but also put it in context of existing treatments for hormone-sensitive tumors. For instance, some discussion on their compatibility with aromatase inhibitors (lines 460-463) should be moved here and expanded. Also, it is important to compare expected adverse effects of STS inhibition with others.

I have now significantly expanded this section to include these comments (Line 109 - 142).

3) in the section 7, it would be important to elaborate why STS did not progress as a single agent and went in the Phase II in combination only. 

This is not completely the case, as the IPET trial did test Irosustat as a single agent in early breast cancer. I have added in a line confirming this (Line 364) and I have added a brief comment on combining arom and STS inhibitors for clinical trial (Line 387 – 389).

Minor concerns:
- compound structures are drawn or copied carelessly, with different and frequently unreadable font size, different line thickness and structure size. Should be redrawn and unified. 

All compounds have now been redrawn using ChemDraw.

- lines 561-565: it is unclear what is meant by "all but Irosustat", probably author meant inverse, but it would be also wrong since there were some other STS inhibitors in Phase I (the cited paper is very old).

This has now been amended (Line 604).

- throughout the text all the greek symbols seen as some strange symbol, alpha sometimes appears as just "a", e.g. in 5a-reductase type-1

These have now been corrected.

- Figure 1 legend: organic anion polypeptides should be  "organic anion transporting polypeptides"

This has now been corrected.

Some noticed typos:

I have now fixed these typos.

Reviewer 2 Report

This review paper describes the past, present, and future of the development of steroid sulfatase (STS) inhibitors, with a particular focus on the research by Prof. Barry Potter, who has been instrumental in many successes in this field. This subject is worthy of review and may contribute to a better understanding of the existing and potential STS inhibitors for further studies. In general, the manuscript is well written and structured, and brings together the scientific advances already achieved and what should be done in this particular field of research. Its publication in Molecules is suggested after minor corrections.

Comments/suggestions:

- Some characters appear deformed/distorted in the manuscript. See lines 52, 53, 123, 157, 162, 180, 181, 182, 212, 214, 333, 386, 434, 444, 530, 532, 536, 580, 584, and 586, and Figure 1 caption.

- Delete “Figure 1” from the top of figure 1.

- Use DHEA-S or DHEAS throughout the manuscript.

- Line 110. Define “RT-PCR” in full.

- Line 299. Indicate in full the meaning of “PBLs”… probably it is “plasmablastic lymphoma”.

- Line 348. Relace “was also determoned” with “was also determined”.

- The chemical structures in Figures 2, 3 and 4 should be improved to standardize the structures size and resolution.

Author Response

I would like to thank the reviewer for their positive comments regarding this manuscript. I have hopefully addressed their comments which I outline below.

Comments/suggestions:

  • Some characters appear deformed/distorted in the manuscript. See lines 52, 53, 123, 157, 162, 180, 181, 182, 212, 214, 333, 386, 434, 444, 530, 532, 536, 580, 584, and 586, and Figure 1 caption.
    • These have now been altered.
  • Delete “Figure 1” from the top of figure 1.
    • Figure 1 title has been deleted.
  • Use DHEA-S or DHEAS throughout the manuscript.
    • DHEA-S has now been standardised throughout manuscript.
  • Line 110. Define “RT-PCR” in full.
    • This has now been defined.
  • Line 299. Indicate in full the meaning of “PBLs”… probably it is “plasmablastic lymphoma”.
    • This has now been defined as Peripheral Blood Lymphocytes"
  • Line 348. Relace “was also determoned” with “was also determined”.
    • This has been changed.
  • The chemical structures in Figures 2, 3 and 4 should be improved to standardize the structures size and resolution.
    • All chemical structures have been now drawn using ChemDraw.